# Theophylline Attenuates the Release of Cardiovascular Disease-Related Triglyceride and Cholesterol by Inhibiting the Activity of Microsomal Triglyceride Transfer Protein in Rat Hepatocytes

**DOI:** 10.3390/biomedicines13112579

**Published:** 2025-10-22

**Authors:** Min-Kyu Park, Hyeonha Jang, Jeong-Soo Bae, Jae-Ho Shin, Hwa-Jin Park

**Affiliations:** 1Department of Applied Biosciences, Kyungpook National University, Daegu 41566, Republic of Korea; pmk601313@knu.ac.kr; 2KNU NGS Core Facility, Kyungpook National University, Daegu 41566, Republic of Korea; 3Microbalance Inc., Daegu 41566, Republic of Korea; 4College of Pharmacy, Kyungpook National University, Daegu 41566, Republic of Korea; jg.heana@gmail.com; 5Cardiovascular Laboratory, Medical Center of Dong-A University, 26 Daesingongwon-ro, Busan 49201, Republic of Korea; hamcholom@hanmail.net; 6Department of Biomedical Laboratory Science, College of Healthcare Medical Science and Engineering, Inje University, 197, Inje-ro, Gimhae 50834, Republic of Korea

**Keywords:** theophylline, rat primary hepatocytes, hypertriglyceridemia, hypercholesterolemia, microsomal triglyceride transfer protein, high-density lipoprotein cholesterol, atherogenic index, cardiovascular disease

## Abstract

**Background/Objectives**: Cardiovascular diseases (CVD) remain the leading cause of diet-related mortality, with hepatic overproduction of very-low-density lipoprotein (VLDL) being a central driver of dyslipidemia. The microsomal triglyceride transfer protein (MTP) is essential for this process, and its activity is negatively regulated by cyclic adenosine monophosphate (cAMP). Theophylline, a methylxanthine found in tea, increases intracellular cAMP. This study aimed to evaluate whether physiologically relevant concentrations of theophylline could beneficially modulate lipoprotein secretion in an ex vivo model of diet-induced MTP activation. **Methods**: Primary hepatocytes were isolated from rats fed a high-fat, high-carbohydrate diet (HFCD). Cells were treated with 100 µM theophylline, and the secretion of triglyceride (TG), total cholesterol (TC), VLDL-cholesterol (VLDL-C), and HDL-cholesterol (HDL-C) was quantified. Hepatocellular MTP activity and atherogenic indices were also assessed. **Results**: Compared to untreated control cells, theophylline treatment significantly reduced the secretion of TG by 6% and TC by 24%. Specifically, VLDL-C secretion decreased by 6%, while HDL-C secretion increased substantially by 93%. These lipid-modulating effects were correlated with a 6.9% reduction in MTP activity. Consequently, significant improvements were observed in the atherogenic indices TG/HDL-C, TC/HDL-C, and the atherogenic index (AI) (*p* < 0.01). **Conclusions**: Our findings demonstrate that physiologically attainable concentrations of theophylline rebalance lipoprotein secretion by suppressing hepatic MTP activity, shifting the lipid profile toward an anti-atherogenic state. These results highlight the potential of theophylline as a functional dietary component for mitigating diet-induced dyslipidemia and reducing cardiovascular risk.

## 1. Introduction

Cardiovascular disease (CVD) remains the leading cause of diet-related mortality, accounting for over 18 million deaths annually [1]. Postprandial dyslipidemia, a hallmark of diet-induced CVD, is driven by the hepatic overproduction of very-low-density lipoprotein (VLDL). VLDL particles, which are rich in triglycerides (TG) and cholesterol esters, are rapidly converted in circulation to low-density lipoprotein (LDL)—the principal atherogenic lipoprotein [2,3,4,5,6,7]. Mounting evidence indicates that aberrant VLDL secretion arises from the upregulation of microsomal triglyceride transfer protein (MTP), a lipid-transfer chaperone located within the endoplasmic reticulum that is essential for the assembly of apolipoprotein B (ApoB) [8]. Nutritional and hormonal signals converge on MTP; for instance, high-fat, high-carbohydrate diets enhance hepatic TG synthesis and intracellular Ca^2+^ levels, both of which proportionally elevate MTP activity and VLDL output, ultimately leading to hypertriglyceridemia and hypercholesterolemia [9,10]. Cyclic adenosine monophosphate (cAMP) functions as a negative regulator; elevated hepatocellular cAMP levels inhibit MTP-mediated TG transfer while concomitantly activating ATP-binding cassette transporter A1 (ABCA1), promoting nascent high-density lipoprotein (HDL) biogenesis [11,12]. These opposing effects position cAMP–MTP signaling as a nutritionally exploitable axis for rebalancing the release of atherogenic versus anti-atherogenic lipoproteins. Methylxanthines serve as key dietary modulators of cAMP, primarily through the non-selective inhibition of phosphodiesterases (PDEs). Theophylline (1,3-dimethylxanthine) is the most abundant methylxanthine found in young tea leaves (Camellia sinensis), with concentrations of 0.1–0.5 mg g^−1^ dry weight; lower levels have been detected in cacao, coffee, and kola nut [13,14]. While synthetic theophylline is clinically used as a bronchodilator, its dietary counterpart, which can easily be extracted from green or partially fermented tea, represents a “clean-label” nutraceutical option. Notably, the dietary intake achievable via tea beverages (<10 µmol/day) is several orders of magnitude lower than the pharmacological doses that have yielded inconsistent lipid outcomes in asthmatic populations [13,15,16]. Whether such physiologically attainable concentrations can beneficially modulate MTP-driven lipoprotein secretion remains untested.

Here, we address this knowledge gap by utilizing primary hepatocytes isolated from rats fed a high-fat, high-carbohydrate diet (HFCD), a validated ex vivo model of diet-induced MTP activation. We hypothesized that tea-equivalent theophylline (i) inhibits MTP activity and the secretion of TG- and cholesterol-rich VLDL and (ii) promotes HDL release, thereby improving atherogenic indices. The present study delineates the direct hepatic actions of theophylline at nutritionally relevant concentrations and provides mechanistic insights regarding its potential application as a functional food component for the management of dyslipidemia.

## 2. Materials and Methods

### 2.1. Materials

Hanks’ balanced salt solution (without calcium chloride, magnesium sulfate, or sodium bicarbonate) for collagenase perfusion and Williams’ medium E (containing L-glutamine but excluding sodium bicarbonate) for hepatocyte culture were purchased from Sigma (St. Louis, MO, USA). Theophylline was also procured from Sigma (St. Louis, MO, USA). TG, total cholesterol (TC), and HDL cholesterol (HDL-C) assay kits were obtained from Nissui Pharmaceutical (Tokyo, Japan). MTP assay kits were acquired from Calbiochem^®^ (an affiliate of Merck KGaA, Darmstadt, Germany). Collagenase (type IV) and additional chemical reagents were secured from Sigma (St. Louis, MO, USA).

### 2.2. Selection and Characteristics of Rat Diets for the Induction of TG and Cholesterol in Hepatocytes

The production of triglycerides (TG) and total cholesterol (TC) is essential for assessing the effect of theophylline on their release by hepatocytes. Microsomal triglyceride transfer protein (MTP), which facilitates the secretion of VLDL containing TG and TC, shows increased activity in proportion to hepatic TG synthesis [7,10,17,18]. To establish this condition, male Sprague–Dawley rats (200–250 g; Samtako Bio, Osan, Republic of Korea) were fed a high-fat, high-carbohydrate diet (HFCD; Table 1) composed of 65.4% fat and carbohydrate (Sam Yang Oil & Fat Feed Co., Ltd., Seoul, Republic of Korea). The HFCD contained soybean oil and lard as fat sources and maltodextrin, cellulose, and sucrose as carbohydrate sources. These nutrients provide precursors for TG and TC synthesis in hepatocytes, thereby inducing obesity, fatty liver, and dyslipidemia, all of which are risk factors for cardiovascular disease (CVD) [10,15,19,20,21,22]. The HFCD (Table 1) was used as a normal control to induce obesity, a risk factor for CVD, in rats [23,24]. In this study, we administered HFCD to rats to increase the production of TG and TC, which are known to contribute to CVD, and investigated the effects of theophylline on lipid profiles. Therefore, the HFCD-administered group served as a control group for theophylline or 10% glucose plus HFCD groups.

To further amplify TG and TC production and thereby MTP activity, 10% glucose was supplemented to the HFCD [10]. Accordingly, hepatocytes isolated from rats fed either HFCD alone or HFCD plus glucose were used to evaluate whether theophylline modulates MTP activity and the secretion of TG and TC.

### 2.3. Experimental Design and Animal Administration

As illustrated in Figure 1, male Sprague–Dawley rats (weight: 200–250 g, 6–7 weeks) secured from Samtako Bio (Osan, Republic of Korea) were categorized into two groups. The first group, designated as the HFCD group (*n* = 5), was provided with HFCD and water *ad libitum.* The second group, referred to as the HFCD plus 10% glucose group (*n* = 5), received the HFCD along with 10% glucose *ad libitum*, with the glucose supplied in place of water. All rats were housed in an environment maintained at a temperature of 24 °C and humidity of 65% under normal light. The duration of rat care spanned 15 days to promote TG synthesis in the liver [25]. To prevent any alterations in TG levels and MTP activity, the rats were not fasted prior to post-mortem analysis [26]. Following the 15-day care period, the rats were anesthetized using 20% urethane before post-mortem procedures. All the animal experiments in this study were conducted in accordance with the Guidelines for Care and Use of Laboratory Animals issued by the Institutional Ethical Animal Care Committee of Kyungpook National University, Korea.

### 2.4. Determination of Theophylline Dose for Measuring Lipid Profiles and MTP Activity

To evaluate the impact of theophylline treatment on CVD-risk lipids—specifically TG, TC, and LDL-C—in the blood of patients with asthma, a previous study administered 300 mg/day of theophylline over a 6-month period, approximately 180 days [16]. This regimen resulted in a cumulative concentration of 300 mmol of theophylline (Mw. 180) (approximately 1.67 mmol/day), which informed our experimental design. We utilized 0.3 μmol (100 μM) of theophylline to investigate the release of CVD risk lipids into the culture medium from rat hepatocytes, representing a theophylline dosage 10–6-fold lower than that (300 mmol) administered to asthmatic patients [16].

The primary metric for evaluating the effect of theophylline on the secretion of TG and TC in hepatocytes is MTP activity. Therefore, we employed 10 mM theophylline, which has previously been utilized to evaluate the secretory activity of insulin in the pancreas, to investigate the influence of theophylline on MTP activity [27].

### 2.5. Rational for Using Hepatocytes to Study Blood Lipid Profiles

Although plasma or serum lipid measurements such as triglycerides, total cholesterol, VLDL, LDL, and HDL are widely used to assess cardiovascular disease (CVD) risk, it is difficult to attribute these changes to specific sources. Circulating lipids originate from multiple pathways, including dietary chylomicrons absorbed from the intestine, VLDL and HDL secreted by the liver, and lipids released from adipose tissue. Because the liver is the predominant source of atherogenic lipids [2,3,4,5,6,7], we investigated the potential anti-CVD mechanism of theophylline by examining hepatocyte-derived lipid secretion. Specifically, we measured VLDL-triglycerides, total cholesterol, and HDL-cholesterol in the culture medium of primary rat hepatocytes isolated from animals fed a high-fat/cholesterol diet (HFCD) or 10% glucose as lipid precursors. These culture medium lipid profiles closely reflect plasma or serum lipid changes and therefore provide mechanistic insights into CVD risk.

### 2.6. Preparation of Rat Hepatocytes and Their Culture

Freshly isolated hepatocytes were prepared from adult male Sprague-Dawley rats (200–250 g) administered with HFCD or HFCD plus 10% glucose for 15 d. Rats were not fasted until the post-mortem to protect any changes on TG level and MTP activity [26]. After 15 d of care, rats were anesthetized with 20% urethane before the post-mortem, and subsequently the hepatocytes were isolated by the collagenase-liver perfusion method of Berry and Friend [28]. Hepatocyte viability was assessed via trypan blue exclusion, yielding an average viability of 95%. Hepatocytes exhibiting 95% viability were washed three times with a washing buffer comprising Williams’ Medium E, 23.8 mM NaHCO_3_, 0.5 mM dexamethasone, and 15 mM 4-(2-hydroxyethyl)-1-piperazineethanesulfonic acid (pH 7.4), supplemented with 1000 U/L penicillin, 1 mg/L streptomycin, and 1 mM insulin. A total of 10^6^ hepatocytes per well, suspended in culture medium (washing buffer supplemented with 10% fetal bovine serum), were placed in culture dishes (Falcon^®^; Becton Dickinson and Company, Franklin Lakes, NJ, USA) and incubated at 37 °C under 5% CO_2_ for 3 h. The monolayer was washed twice with culture medium and supplemented with 3 mL of fresh culture medium. The hepatocytes were subsequently incubated at 37 °C for 6 h, with or without theophylline, after which the incubation was terminated on ice. The culture medium from the hepatocytes in the HFCD group (Group 1) was utilized to measure TG, TC, and HDL-C levels. Hepatocytes prepared from both the HFCD (Group 1) and HFCD plus 10% glucose (Group 2) groups were used to assess MTP activity.

### 2.7. Determination of Lipid Levels in Hepatocyte Culture Medium

Lipid levels in the hepatocyte culture medium were quantified using assay kits from Nissui Pharmaceutical (Tokyo, Japan).

TG levels were measured enzymatically using lipoprotein lipase, glycerol kinase, and glycerol peroxidase. TC was also determined enzymatically using cholesterol esterase, cholesterol oxidase, and peroxidase.

HDL-C levels were assessed enzymatically following the precipitation of VLDL with a complex of sodium phosphotungstic acid and MgCl_2_. VLDL cholesterol (VLDL-C) levels were calculated as follows: VLDL-C = 1/5 TG, as derived from the Friedewald formula: LDL-C = TC − HDL-C − 1/5 TG [29]. The atherogenic index (AI) was calculated using the formula TC − HDL-C/HDL-C. In addition, TG/HDL-C and TC/HDL-C ratios were computed to evaluate the impact of theophylline on CVD risk.

### 2.8. Preparation of Hepatocellular Lysates and Measurement of MTP Activity

To investigate the effects of theophylline on MTP activity, hepatocellular lysates were prepared. Hepatocytes (10^6^ cells/well) were sonicated on ice for two bursts of 10 s each using an ultrasonic homogenizer (Bandelin Sonopuls HD2070; Bandelin Electronic GmbH & Co. KG, Berlin, Germany) set to 30% power.

To measure MTP activity, hepatocellular lysates (200 µg protein) were incubated with gentle agitation at 37 °C for 2 h in the presence of 10 μL of a donor molecule containing a fluorescent neutral lipid and 10 μL of an acceptor molecule, as outlined in the MTP assay kit manual (Calbiochem^®^, Merck KGaA, Darmstadt, Germany). MTP activity was assessed by measuring fluorescence at 535 nm, with excitation at 465 nm, and calculated by subtracting the blank fluorescence value from each sample using a standard curve.

### 2.9. Protein Assay

It is well established that the presence of lipids (e.g., phospholipids, VLDL, LDL) interferes with protein quantification when using the BCA method [30,31,32]. In contrast, chelating agents such as EDTA and EGTA, which are known to interfere with the Lowry method, are absent in hepatocyte culture medium and lysates. Therefore, to avoid lipid-derived interference, we measured protein concentrations using the Lowry method [33]. These measurements were subsequently used to calculate the absolute lipid concentration in the culture medium and the absolute activity of MTP in hepatocytes.

### 2.10. Statistical Analysis

All statistical analyses were performed using GraphPad Prism 8. Prior to data analysis, normality was first checked with the Shapiro–Wilk test and homogeneity of variance with the Brown-Forsythe test. Comparisons of means between two groups were performed using Student’s *t*-test, and comparisons of three or more groups were subjected to one-way analysis of variance (ANOVA) followed by post hoc analysis with Dunnett’s multiple comparison test. All data were expressed as mean ± standard deviation (mean ± SD) of at least three replicate experiments, and were considered statistically significant if *p*-value < 0.05.

## 3. Results

### 3.1. Intake of Feed and Lipids Producing Materials

As shown in Table 2, rats in the HFCD group consumed 20.4 ± 1.5 g of feed per day and gained 3.3 ± 0.1 g of body weight per day. In contrast, rats in the HFCD plus glucose group consumed significantly less feed (14.8 ± 1.7 g/day) but gained greater body weight (3.6 ± 0.9 g/day). Consequently, the feed efficiency ratio (FER) of the HFCD plus glucose group (0.24 ± 0.05) was significantly higher than that of the HFCD group (0.16 ± 0.01), indicating that additional glucose intake (10.6 ± 0.9 g/day) enhanced weight gain relative to feed intake (Table 2). Although rats in the HFCD plus glucose group consumed less dietary fat, oil, and carbohydrates than those in the HFCD group, the additional glucose intake likely served as a major substrate for lipid synthesis.

### 3.2. Effects of Theophylline on TG Release from Hepatocytes into the Culture Medium

Hepatocytes (10^6^ cells/well) derived from rats fed an HFCD released TG (67 ± 1.0 × 10^2^ ng/protein-mg) into the culture medium; nevertheless, theophylline treatment reduced this release to 63 ± 0.8 × 10^2^ ng/protein-mg (Figure 2A). This represents a decrease in culture medium TG levels of approximately 6% compared with that released by the untreated HFCD hepatocytes (Figure 2A). These findings suggest that theophylline inhibits TG release from hepatocytes obtained from rats fed an HFCD as a standard pellet diet.

### 3.3. Effects of Theophylline on TC Release from Hepatocytes into the Culture Medium

As illustrated in Figure 2B, hepatocytes (10^6^ cells/well) from rats on an HFCD released TC (34 ± 1.0 × 10^2^ ng/protein-mg) into the culture medium; nonetheless, theophylline treatment decreased this release to 26 ± 1.0 × 10^2^ ng/protein-mg (Figure 2B). Consequently, these results suggest that theophylline elicits a 24% reduction in TC release from hepatocytes derived from rats fed an HFCD as a standard pellet diet.

### 3.4. Effect of Theophylline on MTP Activity in Hepatocytes

MTP activity serves as an index for evaluating the secretion of TG and TC from hepatocytes [8,34]. Therefore, we examined whether the inhibition of TG and TC secretion by theophylline (Figure 2A,B) correlates with a reduction in MTP activity. As depicted in Figure 3, MTP activity in hepatocytes from rats fed an HFCD supplemented with 10% glucose significantly increased, reaching 27.8 ± 0.8 nM/protein-mg/min, an increase of 25% compared with that observed in hepatocytes on an HFCD alone (221 ± 0.2 nM/protein-mg/min). This implies that the addition of 10% glucose to the HFCD enhances MTP activity in hepatocytes, subsequently increasing the secretion of TG and TC into the culture medium.

To further elucidate the inhibitory effect of theophylline on TG and TC secretion, we investigated its impact on MTP activity in rat hepatocytes administered the HFCD plus 10% glucose, which is known to elevate MTP activity [10]. As illustrated in Figure 3, theophylline reduced the elevated MTP activity from 27.8 ± 0.8 nM/protein-mg/min to 26.0 ± 0.2 nM/protein-mg/min, indicating a 6.9% inhibition of MTP activity induced by the HFCD plus 10% glucose. These findings suggest that the inhibition of TG and TC secretion by theophylline (Figure 2A,B) is at least partially attributable to its inhibitory effect on MTP activity.

### 3.5. Effects of Theophylline on VLDL Release from Hepatocytes into the Culture Medium

TC is secreted from hepatocytes into the bloodstream in the form of TC-poor VLDL and subsequently metabolized into TC-rich LDL. While it is pertinent to investigate theophylline’s effect on CVD risk by measuring TC-rich LDL, it is important to note that the culture medium used in this study for rat primary hepatocytes did not contain the requisite enzymes (e.g., lipoprotein lipase, hepatic lipase, etc.) required to metabolize TC-poor VLDL into TC-rich LDL. Therefore, we assessed theophylline’s potential inhibitory effect on CVD risk by measuring TC-containing VLDL, specifically VLDL-C, which represents the initial lipoprotein that transitions into TC-rich LDL and is exclusively secreted by hepatocytes.

As depicted in Figure 4A, hepatocytes (10^6^ cells/well) derived from HFCD-fed rats released VLDL-C at a rate of 13.4 ± 0.2 × 10^2^ ng/protein-mg into the culture medium. However, theophylline reduced this release to 12.6 ± 0.2 × 10^2^ ng/protein-mg (Figure 4A). These findings indicate that theophylline inhibits VLDL-C release from HFCD-fed rat hepatocytes by 6.0% (Figure 4A), thereby reflecting the compound’s inhibitory effect on TC secretion (Figure 2B).

### 3.6. Theophylline’s Effect on HDL Release from Hepatocytes

Hepatocytes also released HDL independent of MTP activity. In this study, we evaluated the impact of theophylline on HDL release by measuring the concentration of HDL-C in the hepatocyte culture medium. As illustrated in Figure 4B, HFCD-fed rat hepatocytes released HDL-C at a rate of 8 ± 1.0 × 10^2^ ng/protein-mg, which significantly increased to 16 ± 1.0 × 10^2^ ng/protein-mg following theophylline treatment. This suggests that theophylline enhances HDL release from HFCD-fed rat hepatocytes by up to 92.8% (Figure 4B).

### 3.7. Theophylline’s Inhibitory Effect on CVD Risk Lipoproteins and Their Lipids

The impact of theophylline on CVD risk was evaluated using TG, TC, and HDL-C concentrations, which were utilized to calculate the AI, TC/HDL-C ratio, and TG/HDL-C ratio. The average AI derived from HFCD-fed rat hepatocytes was 0.89 (Figure 5A). However, theophylline reduced this value to an average of 0.63, suggesting an inhibitory effect on CVD risk by elevating HDL release compared with VLDL secretion from hepatocytes. This is particularly relevant as TC, used in the AI calculation (TC − HDL-C/HDL-C) is present in both VLDL and HDL. In addition, the TC/HDL-C ratio (1.89 ± 0.1) derived from HFCD-fed rat hepatocytes significantly decreased to 1.63 ± 0.1 with theophylline treatment (Figure 5B). Similarly, the TG/HDL-C ratio decreased from 8.45 ± 0.9 to 3.89 ± 0.3 following theophylline treatment (Figure 5C). These results indicate that theophylline may mitigate CVD risk by promoting HDL-C secretion rather than TG, as evidenced by a 51.2% reduction in the TG/HDL-C ratio (Figure 5C).

## 4. Discussion

The pathological processes underlying the development of CVD are significantly influenced by elevated levels of both TG and TC (i.e., cholesterol esters and free cholesterol) released from hepatocytes. Consequently, numerous researchers have focused on the exploration and development of pharmacological drugs that can effectively lower both TG and TC levels simultaneously, as this reduction is crucial for the treatment and prevention of CVD. Despite several studies suggesting that theophylline exerts no preventive effect on CVD owing to its inability to lower blood levels of TG and TC concurrently [13,15,35], our research indicates otherwise. We found that theophylline inhibits the concurrent secretion of TG and TC from hepatocytes isolated from HFCD-fed rats, typically promoting the increased synthesis of these lipids.

These findings suggest that theophylline inhibits the secretion of TG and TC by suppressing MTP activity in hepatocytes. MTP is responsible for assembling TG into VLDL, along with TC (i.e., free cholesterol and cholesterol esters), phospholipids, and ApoB, for secretion into the blood [5,8]. This assertion is corroborated by our observation wherein theophylline was found to inhibit MTP activity stimulated by the combined intake of the HFCD and 10% glucose, further elevating hepatocellular TG and TC levels beyond those induced by the HFCD alone [10]. Moreover, our results reflect that the secretion of TG and TC increased proportionately with the endogenous production of these lipids in the liver or hepatocytes when an additional 10% glucose was provided alongside the HFCD. This observation aligns with previous studies demonstrating that high-carbohydrate diets can lead to hypertriglyceridemia [36].

Theophylline reduces the secretion of TG from hepatocytes into the culture medium primarily through two mechanisms. First, theophylline increases the intracellular levels of cAMP by inhibiting the activity of cAMP PDE in rat liver [37]. Elevated cAMP levels inhibit the secretory activity of MTP, which is responsible for the export of TG-rich VLDL from hepatocytes [11]. Second, although we did not determine whether theophylline increases cAMP production in primary hepatocytes, it is hypothesized that theophylline, as a PDE inhibitor, enhances the activities of cAMP-dependent adipose triglyceride lipase and hormone-sensitive lipase in hepatocytes, leading to increased TG degradation [38,39,40]. Supporting this, theophylline-based KMUP-1, also a PDE inhibitor, heightens cAMP production, promoting TG degradation in the liver [35]. Furthermore, hepatocellular cAMP is identified as a molecular target for drugs aimed at treating non-alcoholic fatty liver disease induced by HFCD intake or alcoholic fatty liver disease [40].

The mechanism by which theophylline reduces the secretion of TC from hepatocytes into their culture medium can be elucidated as follows: First, theophylline increases intracellular cAMP levels, which may inhibit the VLDL-C secretory activity of MTP. Second, the elevated cAMP levels may enhance the activity of cAMP-dependent cholesteryl ester hydrolase in liver cells, thereby promoting the hydrolysis of cholesterol esters [41,42,43,44]. The precise nature of the relationship between theophylline-induced cAMP elevation and the inhibition of TG-rich VLDL and VLDL-C secretion by MTP remains unclear. It is uncertain whether theophylline-elevated cAMP directly inhibits MTP secretory activity or if the reduction in TG and TC levels results from decreased secretion mediated by elevated cAMP. However, our previous study [11], demonstrated that when rat liver homogenates with lower cAMP and higher TG levels than normal were incubated with dibutyryl-cAMP (db-cAMP), a cAMP analogue, MTP activity decreased in a dose-dependent manner. This finding suggests that increased cAMP in hepatocytes correlates with decreased TG and TC secretion associated with VLDL [45]. Therefore, theophylline, as a PDE inhibitor, presumably suppresses MTP’s secretory activity by promoting cAMP production in hepatocytes, thereby inhibiting TG and TC secretion.

TC comprises both cholesterol esters and free cholesterol; the increased cAMP levels induced by theophylline specifically promote the hydrolysis of cholesterol esters into free cholesterol and free fatty acids. Therefore, the decline in TC secretion observed with theophylline treatment reflects a reduction in the secretion of cholesterol esters. The free fatty acids and free cholesterol generated from the hydrolysis of cholesterol esters in the liver are used to produce ATP and bile acids, which are necessary for metabolic functions [41]. Furthermore, free cholesterol is incorporated into VLDL and HDL and can be secreted from the liver into the culture medium or bloodstream.

Accordingly, monitoring changes in TG and TC levels in the hepatocyte culture medium, as well as VLDL and HDL concentrations in the blood, is instrumental in assessing the effects of theophylline on hypertriglyceridemia and hypercholesterolemia. Among the VLDL and HDL secreted by the liver, VLDL is particularly rich in TG; thus, the reduction in TG levels observed in the culture medium following theophylline treatment indicates that theophylline inhibits the secretion of TG-rich VLDL from the liver.

The TG-rich VLDL secreted into the bloodstream is metabolized into intermediate-density lipoprotein (IDL) by lipoprotein lipase (LPL), which is released from muscle and adipose tissues, facilitating TG decomposition. Subsequently, the TG molecules in IDL are further decomposed into LDL by hepatic lipase secreted from the liver [46]. Notably, during the metabolism of TG-rich VLDL to LDL, TG levels decrease, whereas TC levels increase. This implies that elevated TG-rich VLDL levels in the blood correspond to an increase in LDL, which is characterized by high TC content, ultimately contributing to hypercholesterolemia pathological condition associated with CVD.

Therefore, the reduction in TG levels in the culture medium owing to theophylline suggests that it may inhibit the secretion of TG-rich VLDL, thereby decreasing LDL production and lowering TC levels. Notably, the culture medium used for primary hepatocytes in this experiment lacked LPL, which is necessary for the breakdown of TG in VLDL, IDL, and LDL, resulting in the absence of TC-increasing lipoproteins.

Consequently, the observed decrease in TC in the hepatocyte culture medium following theophylline treatment reflects a reduction in VLDL-C. In fact, theophylline has been revealed to lower VLDL cholesterol levels in the hepatocyte culture medium, suggesting a consequential decline in LDL cholesterol levels, thereby potentially alleviating CVD risk.

In contrast to VLDL and LDL, HDL is a blood lipoprotein associated with an attenuated risk of CVD. Therefore, numerous researchers are investigating substances or drugs that can lower both VLDL and LDL in the blood while simultaneously elevating HDL levels. Theophylline has also been explored for these therapeutic effects; however, it is known not to increase the concentration of HDL in the blood [13,15,16].

Notwithstanding, in this study, we observed that theophylline enhanced the secretion of HDL-C, an indicator of HDL, from primary hepatocytes that had been stimulated to secrete elevated levels of TG and TC following HFCD administration to rats. This finding suggests that theophylline may mitigate HFCD-induced CVD risk. HDL secretion is known to be stimulated by cAMP-phosphorylated ABCA1 in hepatocytes [12,47]. Therefore, theophylline, as a cAMP promoter, plausibly enhances HDL release by phosphorylating ABCA1 in hepatocytes. On evaluating the effects of theophylline on CVD risk indicators, we found it to significantly reduce both the TC/HDL-C and TG/HDL-C ratios in the culture medium of HFCD-fed rat primary hepatocytes.

These findings suggest that theophylline decreases the secretion of TG-rich VLDL containing TC while simultaneously increasing HDL-C secretion, indicating a potential reduction in CVD risk. This conclusion is further supported by the observation that theophylline lowers the AI, a recognized marker of CVD risk.

Importantly, several independent studies have already provided supportive histological evidence regarding the hepatic effects of theophylline. For instance, Dong et al. [48] demonstrated that theophylline reduced hepatic fat accumulation in NAFLD models, Liu et al. [49] reported its protective effects against liver fibrosis and fat deposition in obese mice, and Hussein et al. [50] showed that theophylline alleviated immunologically induced hepatic inflammation. These findings suggest that theophylline exerts consistent beneficial effects on liver pathology, which are in line with our observations.

In summary, our study demonstrates that theophylline, a PDE inhibitor, suppresses MTP activity in hepatocytes, thereby reducing the secretion of VLDL containing triglycerides and cholesterol while promoting HDL secretion. These findings were corroborated in rat hepatocytes, which readily develop hypertriglyceridemia and hypercholesterolemia under HFCD feeding. When considered together with prior reports on the hepatoprotective effects of theophylline, the overall evidence supports the possibility that theophylline may contribute to lowering CVD risk, at least in part, by improving liver function.

## 5. Conclusions

The present ex vivo investigation reveals that nutritionally attainable concentrations of tea-derived theophylline (≤100 µM) directly inhibit MTP activity in rat hepatocytes primed with an HFCD. This inhibition curtails the secretion of atherogenic TG, TC, and VLDL-C while simultaneously doubling HDL-C output, thereby improving the TC/HDL-C and TG/HDL-C ratios as well as the AI. Collectively, these data position theophylline as a promising modulator of dyslipidemia and a potential adjunct for CVD risk reduction. Further confirmation through in vivo models and human dietary intervention trials is warranted.

The beneficial effects of theophylline observed in this study were achieved without evidence of cytotoxicity or alterations in cell viability, underscoring the specificity of its modulatory action on lipid metabolism. Considering that theophylline is naturally abundant in common dietary sources, such as green tea, cacao, and kola nuts, its regular consumption may provide a practical and accessible means of managing dyslipidemia and associated cardiovascular risks. Furthermore, the physiologically relevant concentrations employed in this study underscore the translational potential of these findings to everyday dietary practices and the development of nutraceutical products.

While these ex vivo results are promising, we recognize the limitations of directly translating findings from cell-based models to clinical practice. One limitation is the relatively small sample size of five rats per group. This was chosen in accordance with the 3Rs principle, and the study was sufficiently powered to detect highly significant effects (e.g., *p* < 0.01 and *p* < 0.0001). Additionally, this study focused on a model of diet-induced dyslipidemia and therefore did not include a normal diet control group. This limits the generalization of our findings to physiological conditions in a healthy state. Nevertheless, future studies with larger animal cohorts would strengthen the generalizability of our conclusions. Furthermore, we exclusively used male rats to minimize metabolic variability arising from the female estrous cycle, allowing for a clearer interpretation of theophylline’s direct effects. As sex-based differences in lipid metabolism are well-documented, investigating these effects in female animals is a valuable direction for future research. Therefore, future research should extend these findings to animal models and human clinical trials to rigorously evaluate theophylline’s effectiveness, optimal dosing regimens, and long-term safety across diverse populations. Confirmation through comprehensive in vivo experiments and well-controlled human dietary intervention studies will be critical to fully validate theophylline’s role as an adjunctive therapeutic strategy for CVD risk reduction and its incorporation into dietary guidelines and public health recommendations.

## Figures and Tables

**Figure 1 biomedicines-13-02579-f001:**
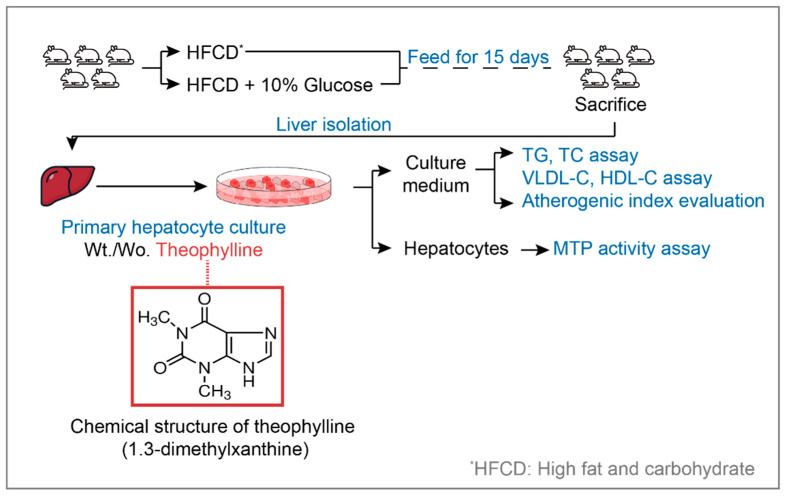
**Study design to investigate the effects of theophylline on CVD risk.** TG: triglyceride, TC: total cholesterol, HDL-C: high density lipoprotein-cholesterol, MTP: microsomal triglyceride transfer protein, Wt: with, Wo: without.

**Figure 2 biomedicines-13-02579-f002:**
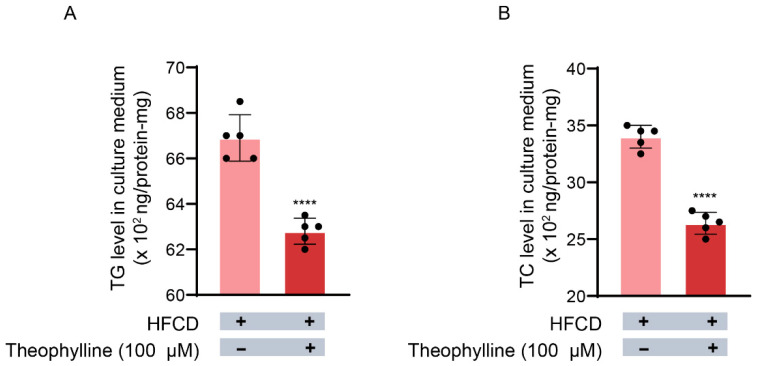
**Effect of theophylline on the release of triglyceride and total cholesterol (TC) into hepatocytes culture medium.** The levels of TG and TC were determined as described in Materials and Methods. (**A**) Level of TG. (**B**) Level of TC. TG, triglyceride; TC, total cholesterol. HFCD, high fat and carbohydrate diets. Data are given as means ± SD., *n* = 5. **** *p* < 0.0001.

**Figure 3 biomedicines-13-02579-f003:**
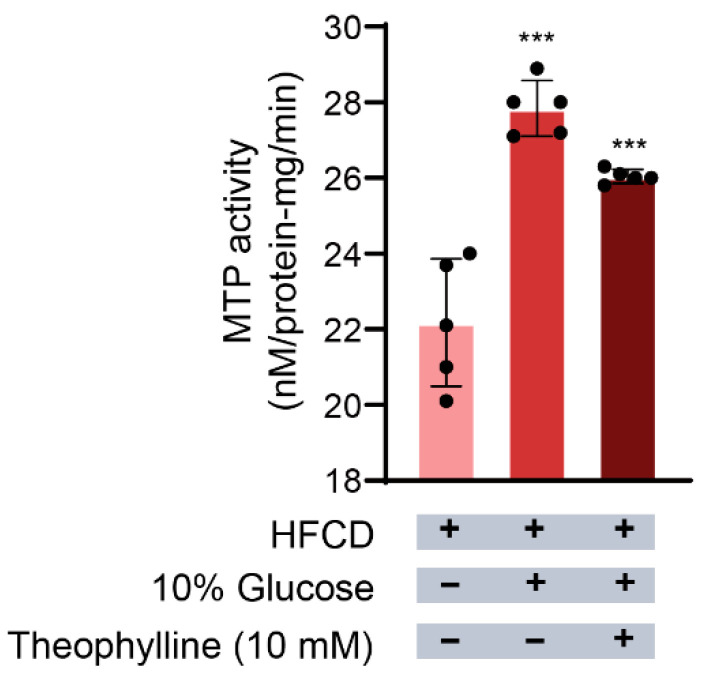
**Effects of theophylline on MTP activity in hepatocytes.** MTP activity was determined as described in Materials and Methods. HFCD, high fat and carbohydrate diets. MTP, microsomal triglyceride transfer protein. Data are given as means ± SD., *n* = 5. *** *p* < 0.001.

**Figure 4 biomedicines-13-02579-f004:**
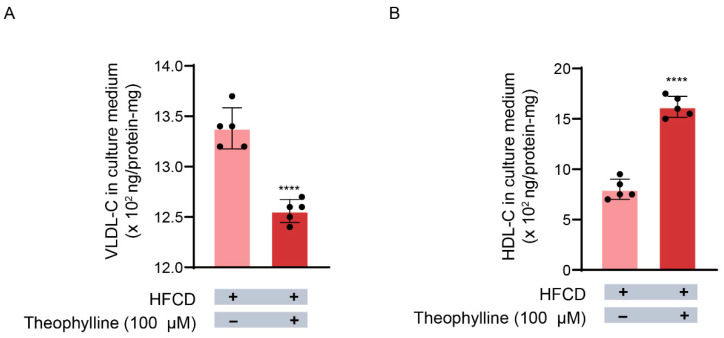
**Effects of theophylline on the release of VLDL-C and HDL-C into hepatocytes culture medium.** The level of VLDL-C was calculated and the level of HDL-C was determined as described in Materials and Methods. (**A**) Level of VLDL-C. (**B**) Level of HDL-C. HFCD, high fat and carbohydrate diets; VLDL-C, very low density lipoprotein; HDL-C, high density lipoprotein. Data are given as means ± SD., *n* = 5. **** *p* < 0.0001.

**Figure 5 biomedicines-13-02579-f005:**
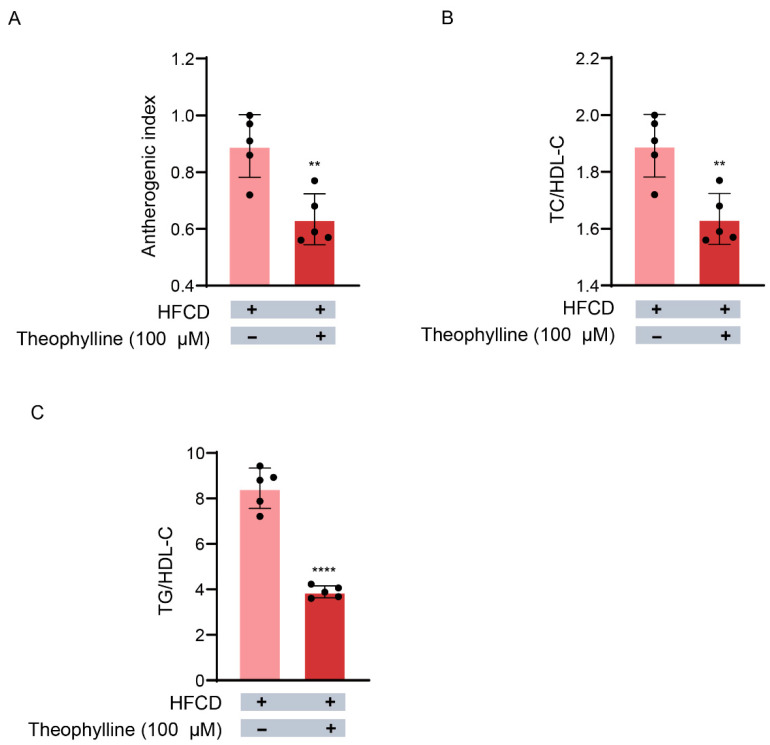
**Effects of theophylline on indexes of CVD risk.** Atherogenic index (AI), TC/HDL-C ratio and TG/HDL-C were calculated with the levels of TG, TC and HDL-C. (**A**) Atherogenic index. (**B**) TC/HDL-C ratio. (**C**) TG/HDL-C. HFCD, high fat carbohydrates diets. Data are given as means ± SD., *n* = 5. ** *p*< 0.01 and **** *p* < 0.0001.

**Table 1 biomedicines-13-02579-t001:** High-fats and carbohydrates diet composition (Sam Yang Oil & Fat Feed Co., Republic of Korea).

Ingredient		Contents	
		(g/kg)	(%)
Fat and oil	Soybean oil	25	65.4
	Lard	245	
	Subtotal	270	
Carbohydrate	Maltodextrin	125	
	Sucrose	68.8	
	Cellulose	50	
	Subtotal	243.8	
Protein and amino acid	Casein	200	25.8
	L-Cystine	3	
Mineral and electrolytes	Mineral mix	10	7.3
	Dicalcium	13	
	Phosphate	5.5	
	Calcium	16.5	
	Carbonate	10	
	Potassium citrate	2	
Vitamins	Vitamin mix	10	1.5
	Choline bitartrate	2	
Total		785.8	100

Fat and oil, and carbohydrates representing HFCD are shaded.

**Table 2 biomedicines-13-02579-t002:** Daily feed intake and lipid-producing substrate concentration in rats.

Animal Group	Feed Intake (g/d)	Body Weight Gain (g/d)	FER	Intake of Lipid-Producing Materials of Feed (g/d)			
				Fat and Oil (270 g/kg-HFCD)	Carbohydrates (243.8 g/kg-HFCD)	Glucose (g/d)	Total
HFCD	^(1)^ 20.4 ± 1.5	3.3 ± 0.1	0.16 ± 0.1	^(3)^ 5.51 ± 0.41	^(5)^ 4.97 ± 0.41	-	10.48 ± 0.8
HFCD + glucose	^(2)^ 14.8 ± 1.7 ***	3.6 ± 0.9 *	0.24 ± 0.5	^(4)^ 4.00 ± 0.46	^(6)^ 3.60 ± 0.41 **	10.6 ± 0.9	18.2 ± 1.7 ***

* *p* < 0.05, ** *p* < 0.01, *** *p* < 0.001. Data are given as means ± S.D. (*n* = 5). HFCD, high fat and carbohydrate. FER, feed efficiency ratio. FER = body weight gain (g/d)/feed intake (g/d). (3) = (1) × 270 g/kg-HFCD. (4) = (2) × 270 g/kg-HFCD. (5) = (1) × 243.8 g/kg-HFCD. (6) = (2) × 243.8 g/kg-HFCD. Fat and oil (270 g/kg-HFCD) and carbohydrates (243.8 g/kg-HFCD) were derived from Table 1.

## Data Availability

The raw data supporting the conclusions of this article will be made available by the authors on request.

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
