# Peer review of "Theophylline Attenuates the Release of Cardiovascular Disease-Related Triglyceride and Cholesterol by Inhibiting the Activity of Microsomal Triglyceride Transfer Protein in Rat Hepatocytes"

_biomedicines, 2025, doi:10.3390/biomedicines13112579_

Round 1

Reviewer 1 Report

Comments and Suggestions for Authors

The authors presented an interesting manuscript on the potential of theophylline for alleviating diet-induced dyslipidemia and reducing cardiovascular risk. However, the presented study displays some issues that should be considered:

Section 2.3.

  1. Please add the age of the rats.
  2. Why only 5 rats per group? Authors should add a justified explanation, since this is a major concern. . At least 6 animals per group should be included. Why only male rats?
  3. Could authors include the BWG results?

Section 3.9.

  1. What statistics have been used?
  2. Furthermore, could the author provide a more detailed explanation of why the Student’s t-test was used? What was the distribution of variables?

Author Response

Comments 1: The authors presented an interesting manuscript on the potential of theophylline for alleviating diet-induced dyslipidemia and reducing cardiovascular risk. However, the presented study displays some issues that should be considered:

Answer) Thank you very much for your time and for the thoughtful review of our manuscript, "The potential of theophylline for alleviating diet-induced dyslipidemia and reducing cardiovascular risk." We are grateful for your insightful feedback and constructive comments. We have carefully considered the issues you raised and appreciate the opportunity to improve our work. We acknowledge the concerns you have highlighted and will address each point thoroughly in a revised version of the manuscript. Your feedback has been invaluable in helping us to strengthen our study and clarify our presentation. We will prepare a point-by-point response to your comments and revise the manuscript accordingly. We are confident that these changes will address your concerns and significantly improve the quality of our paper. Thank you once again for your valuable contribution to the peer-review process.

Comments 2: Section 2.3. Please add the age of the rats.

Answer) Thank you for pointing out this omission. In the revised manuscript, we will specify the age of the rats. The male Sprague-Dawley rats used in our study had a starting weight of 200–250 g, which corresponds to an age of approximately 6-7 weeks. We will add this information to the "Experimental design and animal administration" subsection (2.3) of the "Materials and Methods" to provide a more complete description of the animals (line 122).

Comments 3: Why only 5 rats per group? Authors should add a justified explanation, since this is a major concern. . At least 6 animals per group should be included. Why only male rats?

Answer) 1) Sample size (n=5): In line with the 3Rs principle (Reduction, Refinement, Replacement) for animal experimentation, we chose a sample size of five rats per group, which our preliminary studies and the existing literature indicated would be sufficient to detect statistically significant differences. Indeed, our results demonstrated highly significant effects (e.g., p<0.01 and p<0.0001), supporting the adequacy of this sample size for the primary outcomes. However, we agree that a larger sample size would further strengthen our conclusions, and we will add a statement acknowledging this as a limitation in the "Conclusion" section of the revised manuscript (line 479-487).

2) Use of Main Rats: We used only male Sprague-Dawley rats to minimize potential metabolic variability arising from hormonal fluctuations associated with the female estrous cycle. As our study focuses on the sensitive mechanisms of lipid metabolism and lipoprotein secretion, using a single sex provides a more homogenous model, allowing for a clearer interpretation of the direct effects of theophylline. We acknowledge the importance of investigating these effects in female animals and will add a note in the "Conclusion" section suggesting this as a valuable direction for future research (line 479-487).

Comments 4: Could authors include the BWG results?

Answer) Thank you very much for your comments and attention. This is an excellent suggestion. While the primary intervention with theophylline was conducted ex vivo on isolated hepatocytes, providing data on the animals' physiological state after the 15-day feeding period would provide valuable context. During the diet induction period, food intake (g/d) and body weight gain (BWG, g/d) in the HFCD and HFCD + 10% glucose groups were monitored to calculate the feed efficiency ratio (FER) to present the efficiency of the feed during the diet intake period. In addition, the amounts of lipid-producing materials ingested by the rats in the HFCD and HFCD plus 10% glucose groups were provided to facilitate understanding of the effect of theophylline on the changes in lipid profiles. These are presented in Table 2 in addition to the “Results” section (line 229-238) of the revised manuscript.

Comments 5: Section 3.9. What statistics have been used? Furthermore, could the author provide a more detailed explanation of why the Student’s t-test was used? What was the distribution of variables?

 Answer) Thank you for your comments and attention. As pointed out by the reviewer, we have clarified the statistical techniques used in this study. We used Student's t-test for mean comparisons between two groups and One-way ANOVA for mean comparisons between three or more groups. If statistically significant differences were observed as a result of ANOVA analysis, we performed post hoc tests using Dunnett's test to specifically determine which groups differed. We're responding to your request to add a more detailed explanation of why you used Student's t-test. Student's t-test is a parametric statistics technique that has high reliability and power when the data satisfy two key assumptions: normality and homogeneity of variances. Therefore, we performed the Shapiro-Wilk test and Brown-Forsythe test before applying the t-test. The results showed that all the data we analyzed followed a normal distribution and that the variances between the groups were homogeneous. Since we confirmed that both of these assumptions were met, we decided to use the Student's t-test, which is the most appropriate and reliable method for comparing mean differences between two groups. In response to the reviewer's comments, we have made the following detailed revisions to Section 3.9. Statistical Analysis in the manuscript (line 220 – 227).

Reviewer 2 Report

Comments and Suggestions for Authors

The study presents findings on the potential role of theophylline in modulating lipid metabolism and cardiovascular disease (CVD) risk by targeting microsomal triglyceride transfer protein (MTP) in rat hepatocytes. However, several key concerns and clarifications are required before the manuscript can be considered for publication. Below are my detailed comments:

  1. The authors utilized the Lowry method for protein quantification. While the Lowry method is a classic approach, it is prone to interference from certain substances (e.g., detergents, ions) and is less stable compared to modern methods like the BCA assay. The authors should justify their choice of the Lowry method over BCA. Specifically, they should explain whether the experimental conditions necessitated the use of Lowry (e.g., compatibility with specific buffers or sample types) or if the advantages of BCA (e.g., greater stability, fewer interferences) were considered but deemed unsuitable.
  2. In Figure 2 and subsequent experiments, the authors only compared theophylline-treated groups with high-fat, high-carbohydrate diet (HFCD) groups. A normal control group (e.g., rats fed a standard diet) is entirely missing. Without a normal control, it is difficult to assess whether the observed effects of theophylline are specific to HFCD-induced dyslipidemia or represent broader metabolic changes. The authors should include data from a normal control group to strengthen the validity of their conclusions.
  3. The study focuses on isolated hepatocytes rather than directly measuring lipid profiles or CVD risk markers in live animals. While hepatocyte cultures provide insights into cellular mechanisms, they do not fully reflect systemic lipid metabolism or CVD risk in vivo. The authors should explain why they chose to prioritize cell-based assays over whole-animal measurements. Additionally, they should discuss the limitations of using isolated hepatocytes and how these findings might translate to in vivo conditions.
  4. While the study identifies MTP as a target of theophylline, the molecular signaling pathways underlying its effects remain unclear. For example, how does theophylline influence cAMP levels and MTP activity at the molecular level? Are there specific protein-protein interactions or post-translational modifications involved? The authors should supplement their findings with experiments elucidating the mechanistic basis of theophylline’s action (e.g., Western blotting for phosphorylated proteins, cAMP quantification).
  5. The manuscript does not include histological assessments of animal tissues (e.g., liver sections to evaluate steatosis or inflammation). Such analyses would provide critical evidence of theophylline’s effects on lipid deposition and tissue pathology. Additionally, the authors rely solely on biochemical markers of CVD risk. While these markers are valuable, they should be complemented with histological data to offer a more comprehensive assessment of CVD risk mitigation.

Author Response

[Reviewer 2]

The study presents findings on the potential role of theophylline in modulating lipid metabolism and cardiovascular disease (CVD) risk by targeting microsomal triglyceride transfer protein (MTP) in rat hepatocytes. However, several key concerns and clarifications are required before the manuscript can be considered for publication. Below are my detailed comments:

 Answer) Thank you for your thorough and insightful review of our manuscript. We sincerely appreciate the time and expertise you dedicated to providing feedback. Your comments are invaluable and have helped us significantly improve the clarity, rigor, and overall quality of our paper. We have carefully considered all the points raised and have revised the manuscript accordingly. Below, we provide a point-by-point response to each of your comments, detailing the changes we have made. We believe the revisions have strengthened our manuscript, and we hope you will now find it suitable for publication in Biomedicines. We look forward to your feedback. Thank you again for your constructive guidance.

Comments 1: The authors utilized the Lowry method for protein quantification. While the Lowry method is a classic approach, it is prone to interference from certain substances (e.g., detergents, ions) and is less stable compared to modern methods like the BCA assay. The authors should justify their choice of the Lowry method over BCA. Specifically, they should explain whether the experimental conditions necessitated the use of Lowry (e.g., compatibility with specific buffers or sample types) or if the advantages of BCA (e.g., greater stability, fewer interferences) were considered but deemed unsuitable.

Answer) We appreciate the reviewer’s important observation. The choice of protein quantification method depends on the characteristics of the sample and the purpose of the experiment. We carefully selected the protein measurement method to obtain the absolute amount of lipids in the hepatocyte culture medium or the absolute activity of MTP of the hepatocytes. As you described, bicinchoninic acid (BCA) is widely used for protein quantification analysis. However, (1) it is well known that the presence of lipids (phospholipids, VLDL, LDL, HDL, etc.) in the sample interferes with protein measurement when measuring protein with BCA (Kessler and Fanestil, 1986; Morton and Evans, 1992; Instruction PierceⓇ BCA Protein Assay Kit, 23225, 23227, Thermo Scientific). (2) EDTA and EGTA, which interfere with protein measurement by the Lowry method, do not exist in hepatocyte culture medium or hepatocyte lysates. Therefore, we measured protein using the Lowry method to exclude interference from lipids. These were attached in “section 2.9. Protein assay’ of the revised manuscript (line 209-218).

References:

  1. Kessler, R. J.and Fanestil D.D (1986) Interference by lipids in the determination of protein using bicinchoninic acid. Anal. Biochem. 159: 138-142.
  2. Morton, R.E. and Evans T.A. (1992) Modification of the bicinchoninic acid protein assay to eliminate lipid interference in determining lipoprotein protein content. Anal. Biochem. 204:332-34.
  • Thermo Scientific, Instruction PierceⓇ BCA protein Assay Kit, 23225, 23227

These references are attached as 30, 31 and 32 respectively in the References section of the revised manuscript.

Comments 2: In Figure 2 and subsequent experiments, the authors only compared theophylline-treated groups with high-fat, high-carbohydrate diet (HFCD) groups. A normal control group (e.g., rats fed a standard diet) is entirely missing. Without a normal control, it is difficult to assess whether the observed effects of theophylline are specific to HFCD-induced dyslipidemia or represent broader metabolic changes. The authors should include data from a normal control group to strengthen the validity of their conclusions.

Answer) We appreciate the reviewer's valuable feedback. The HFCD (Table 1) was used as a normal control to induce obesity, a risk factor for CVD, in rats (Kim et al., 2016; Kim and Lee, 2014). In this study, we administered HFCD to rats to increase the production of triglycerides and cholesterol, which are known to contribute to CVD, and investigated the effects of theophylline on lipid profiles. Therefore, the HFCD-administered group served as a control group for theophylline or 10% glucose plus HFCD groups. Related contents were attached in “section 2.2” of the revised manuscript (line 110-115)

References:

  1. Kim et al. (2016) Effects of exercise and L-arginine intake on inflammation in aorta of high-fat diet induced obse rats. J. Exerc Nutrition Biochem. 20,036-040.
  2. Kim and Lee (2014) Exercise training suppresses vascular fibrosis in aging obesity induced rats. J. Exerc Nutrition Biochem. 18, 175-180.

These references are attached as 23 and 24 respectively in the References section of the revised manuscript.

Comments 3: The study focuses on isolated hepatocytes rather than directly measuring lipid profiles or CVD risk markers in live animals. While hepatocyte cultures provide insights into cellular mechanisms, they do not fully reflect systemic lipid metabolism or CVD risk in vivo. The authors should explain why they chose to prioritize cell-based assays over whole-animal measurements. Additionally, they should discuss the limitations of using isolated hepatocytes and how these findings might translate to in vivo conditions.

Answer) Thank you for this valuable observation. Although plasma or serum lipid measurements such as triglycerides, total cholesterol, VLDL, LDL, and HDL are commonly used to assess the risk of cardiovascular disease (CVD), it is difficult to attribute these changes to specific sources. This is because circulating lipids originate from multiple pathways, including dietary chylomicron-derived lipids absorbed from the intestine, VLDL- and HDL-derived lipids secreted by the liver, and lipids released from adipose tissue. Since the liver is the predominant source of atherogenic lipids (Higgins and Hutson, 1984; Gibbons, 1990; Vance and Vance, 1990; Griffin and Zampelas, 1995; Benoist et al., 1996; Gordon and Jamil, 2000), we investigated the potential anti-CVD mechanism of theophylline by examining VLDL-derived triglycerides, total cholesterol, and HDL-derived cholesterol (HDL-C) secreted into the culture medium of primary rat hepatocytes isolated from animals fed a high-fat/cholesterol diet (HFCD) or 10% glucose as lipid precursors. Changes in lipid profiles in the hepatocyte culture medium, which closely reflect those observed in plasma or serum, can therefore provide mechanistic insights into CVD risk. These contents were attached in line 150-162 as section “2.5. Rational for using hepatocytes to study blood lipid profiles” of the revised manuscript.

Comments 4: While the study identifies MTP as a target of theophylline, the molecular signaling pathways underlying its effects remain unclear. For example, how does theophylline influence cAMP levels and MTP activity at the molecular level? Are there specific protein-protein interactions or post-translational modifications involved? The authors should supplement their findings with experiments elucidating the mechanistic basis of theophylline’s action (e.g., Western blotting for phosphorylated proteins, cAMP quantification).

Answer) Unfortunately, the current study cannot provide definitive evidence that theophylline increases hepatocellular cAMP levels. However, it can be argued that theophylline can inhibit the activity of MTP in hepatocytes by increasing cAMP levels. The following findings support this: 1) Theophylline increases cAMP levels in the liver (Hickie et al., 1975). 2) cAMP inhibits MTP activity in hepatocytes (Cho et al., 2009). These details are described in lines 410–416 of the revised manuscript.

References:

  1. Hickie, R.A.; Walker, C.M.; Datta, A. Increased activity of low-Km cyclic adenosine 3′:5′-monophosphate phosphodiesterase in plasma membranes of Morris hepatoma 5123tc (h). Cancer Res. 1975, 35, 601-605.
  2. Cho, H.-J.; Kim, H.-S.; Yu, Y.-B.; Kang, H.-C.; Lee, D.-H.; Rhee, M.-H.; Cho, J.-Y.; Park, H.-J. The opposite correlation between calcium ion and cyclic-AMP regarding the activation of microsomal triglyceride transfer protein in rat liver. BMB Rep. 2009, 42, 642-647. https://doi: 10.5483/bmbrep.2009.42.10.642

These references are attached as 11 and 37 respectively in the References section of the revised manuscript.

Comments 5: The manuscript does not include histological assessments of animal tissues (e.g., liver sections to evaluate steatosis or inflammation). Such analyses would provide critical evidence of theophylline’s effects on lipid deposition and tissue pathology. Additionally, the authors rely solely on biochemical markers of CVD risk. While these markers are valuable, they should be complemented with histological data to offer a more comprehensive assessment of CVD risk mitigation.

Answer) Thank you very much for your comments and attention.

   We sincerely appreciate the reviewer’s insightful comment. We fully agree that histological evaluation of liver tissues would provide stronger evidence for clarifying the CVD risk–reducing potential of theophylline. In fact, such analysis could offer direct morphological support for the biochemical and molecular findings. However, as described in Figure 1. Study design to investigate the effects of theophylline on CVD risk, the primary objective of the present study was to evaluate CVD risk by assessing lipid profiles and MTP activity in primary hepatocytes isolated from HFCD- and HFCD plus glucose–administered rats. Therefore, histological assessments were not included within the scope of the current experimental design.

Importantly, several independent studies have already provided supportive histological evidence regarding the hepatic effects of theophylline. For instance, Dong et al. (2025) demonstrated that theophylline reduced hepatic fat accumulation in NAFLD models, Liu et al. (2022) reported its protective effects against liver fibrosis and fat deposition in obese mice, and Hussein et al. (2015) showed that theophylline alleviated immunologically induced hepatic inflammation. These findings suggest that theophylline has consistent beneficial effects on liver pathology, which are in line with our observations.

When considered together with our results showing that theophylline suppressed the secretion of triglycerides and cholesterol by reducing hepatic MTP activity, the overall evidence supports the possibility that theophylline may lower CVD risk, at least in part, by improving liver function. Nonetheless, we fully acknowledge that additional studies incorporating direct histological validation will be essential to strengthen this conclusion and further elucidate the underlying mechanisms. These contents were attached to line 447-453, and 458-460 of the revised manuscript.

References:

  1. Dong, J.; Du, X.; Yang, R.; Shan, L.; Lu, X.; Shen, Y.; Li, Y.; Duan, S.; Du, Z.; Fueng, J.; fang, C. Tramnscriptomic analysis revels the mechanisms underlying the differential effects of caffeine, theophylline, and theobromine in regulating hepatic fat accumulation. & Function. 2025., 16 (Suppl 1) Doi: 10.1039/d4fo04001e
  2. Liu, T-T.Liu, X-T.; Huang, G-L.; Liu, L.; Chen, Q-X.; Wang, Q. Theophylline extracyed from Fu Brick Tea affects the metabolism of preadipocytes and body fat in mice as a pancreatic lipase inhibitor. Int. J. Mol. Sci. 2022, 23, 2525. http://doi.org/10.3390/ijms23052525
  • Hussein, R.M.; Elsirafy, M.; Wahba, Y.S.; Kawy, H.S.A.; Hasanin, A.H.; Galal, G. Theophylline, an ulti-faceted effects: Its potential benefits in immunological liver injury in rats. Life Sci. 2015, 136-100-7. doi: 10.1016/j.lfs.2015.06.028

These references are attached as 48, 49, and 50 respectively in the References section of the revised manuscript.

Round 2

Reviewer 1 Report

Comments and Suggestions for Authors The authors responded adequately to all my suggestions, and the manuscript is now suitable for publication. Best regards.

Author Response

Comments 1: The authors responded adequately to all my suggestions, and the manuscript is now suitable for publication. Best regards.

Answer) Thank you very much for your positive assessment and for recommending our manuscript for publication.

We sincerely appreciate your time and constructive comments throughout the review process, which have significantly improved the quality of our manuscript. We are truly grateful for your guidance and support.
